# Object Scene Representation Transformer

**Mehdi S. M. Sajjadi, Daniel Duckworth**[*], **Aravindh Mahendran**[*]**, Sjoerd van Steenkiste**[*]**,
Filip Pavetić, Mario Lučić, Leonidas J. Guibas, Klaus Greff, Thomas Kipf**[*]

Google Research

## Abstract

A compositional understanding of the world in terms of objects and their geometry in 3D space is considered a cornerstone of human cognition. Facilitating the learning of such a representation in neural networks holds promise for substantially improving labeled data efficiency. As a key step in this direction, we make progress on the problem of learning 3D-consistent decompositions of complex scenes into individual objects in an unsupervised fashion. We introduce *Object Scene Representation Transformer (OSRT)*, a 3D-centric model in which individual object representations naturally emerge through novel view synthesis. OSRT scales to significantly more complex scenes with larger diversity of objects and backgrounds than existing methods. At the same time, it is multiple orders of magnitude faster at compositional rendering thanks to its light field parametrization and the novel *Slot Mixer* decoder. We believe this work will not only accelerate future architecture exploration and scaling efforts, but it will also serve as a useful tool for both object-centric as well as neural scene representation learning communities.

## 1   Introduction

As humans, we interact with a physical world that is composed of macroscopic objects[1] situated in 3D environments. The development of an object-centric, geometric understanding of the world is considered a cornerstone of human cognition [32]: we perceive scenes in terms of discrete objects and their parts, and our understanding of 3D scene geometry is essential for reasoning about relations between objects and for skillfully interacting with them.

Replicating these capabilities in machine learning models has been a major focus in computer vision and related fields [12, 22, 34], yet the classical paradigm of supervised learning poses several challenges: explicit supervision requires carefully annotated data at a large scale, and is subject to obstacles such as rare or novel object categories. Further, obtaining accurate ground-truth 3D scene and object geometry is challenging and expensive. Learning about compositional geometry merely by observing scenes and occasionally interacting with them—in the simplest case by moving a camera through a scene—without relying on direct supervision is an attractive alternative.

As objects in the physical world are situated in 3D space, there has been a growing interest in combining recent advances in 3D neural rendering [25] and representation learning [7, 20] with object-centric inductive biases to jointly learn to represent objects and their 3D geometry without direct supervision [26, 33, 40]. A particularly promising setting for learning both about objects and 3D scene geometry from RGB supervision alone is that of *novel view synthesis (NVS)*, where the task is to predict a scene's appearance from unobserved view points. This task not only encourages a model to learn a geometrically-consistent representation of a scene, but has the potential to serve as an additional inductive bias for discovering objects without supervision.

---

Website: osrt-paper.github.io. Correspondence: osrt@msajjadi.com. [*]Equal technical contribution.
[1]We use the term 'object' in a very broad sense, capturing other physical entities such as embodied agents.

36th Conference on Neural Information Processing Systems (NeurIPS 2022).

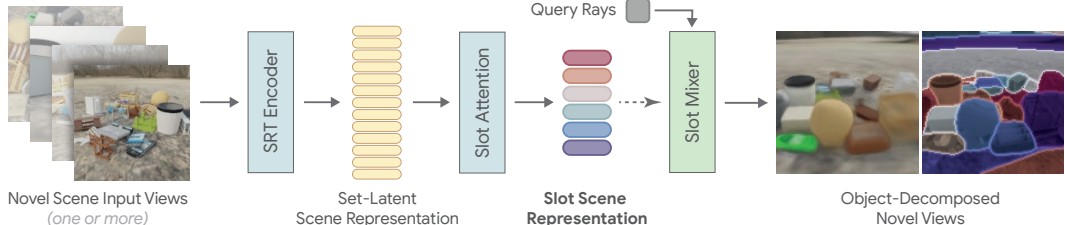

Figure 1: **OSRT overview** – A set of input views of a novel scene are processed by the *SRT Encoder*, yielding the *Set-Latent Scene Representation (SLSR)*. The *Slot Attention* module converts the SLSR into the object-centric *Slot Scene Representation*. Finally, arbitrary views with 3D-consistent object instance decompositions are efficiently rendered by the novel *Slot Mixer*. The entire model is trained end-to-end with an L2 loss and no additional regularizers. Details in Sec. 2.

Prior methods combining object-centric inductive biases with 3D rendering techniques for NVS [26, 33, 40] however fail to generalize to scenes of high visual complexity and face a significant shortcoming in terms of computational cost and memory requirements: common object-centric models decode each object independently, adding a significant multiplicative factor to the already expensive volumetric rendering procedure which requires hundreds of decoding steps. This requirement of executing thousands of decoding passes for each rendered pixel is a major limitation that prohibits scaling this class of methods to more powerful models and thus to more complex scenes.

In this work, we propose *Object Scene Representation Transformer (OSRT)*, an end-to-end model for object-centric 3D scene representation learning. The integrated Slot Attention [23] module allows it to learn a scene representation that is decomposed into *slots*, each representing an object or part of the background. OSRT is based on SRT, utilizing a light field parameterization of space to predict pixel values directly from a latent scene representation in a single forward pass. Instead of rendering each slot independently, we introduce the novel *Slot Mixer*, a highly-efficient object-aware decoder that requires just a single forward pass per rendered pixel, irrespective of the number of slots in the model. In summary, OSRT allows for highly scalable learning of object-centric 3D scene representations without supervision.

Our core contributions are as follows:

- We present OSRT, a model for 3D-centric representation learning, enabling efficient and scalable object discovery in 3D scenes from RGB supervision. The model is trained purely with the simple L2 loss and does not necessitate further regularizers or additional knowledge such as depth maps [33] or explicit background handling [40].

- As part of OSRT, we propose the novel *Slot Mixer*, a highly efficient object-aware decoder that scales to large numbers of objects in a scene with little computational overhead.

- We study several properties of the proposed method, including robustness studies and advantages of using NVS to facilitate scene decomposition in complex datasets.

- In a range of experiments from easier previously proposed, to complex many-object scenes, we demonstrate that OSRT achieves state-of-the-art object decomposition, outperforming prior methods both quantitatively and in terms of efficiency.

## 2 Method

We begin this section with a description of the proposed *Object Scene Representation Transformer (OSRT)* shown in Fig. 1 and its novel *Slot Mixer* decoder, and conclude with a discussion of possible alternative design choices.

### 2.1 Novel view synthesis

The starting point for our investigations is the *Scene Representation Transformer (SRT)* [29] as a geometry-free novel view synthesis (NVS) backbone that provides instant novel-scene generalization and scalability to complex datasets. SRT is based on an encoder-decoder architecture.

A data point consists of a set of RGB *input images* $\{I_i \in \mathbb{R}^{H \times W \times 3}\}$ from the same scene.[2] A convolutional network CNN independently encodes each image into a feature map, all of which are finally flattened and combined into a single set of tokens. An *Encoder Transformer* $\mathcal{E}$ then performs self-attention on this feature set, ultimately yielding the *Set-Latent Scene Representation (SLSR)*

$$\{\mathbf{z}_j \in \mathbb{R}^d\} = \mathcal{E}_\theta(\{\text{CNN}_\theta(\mathbf{I}_i)\}). \tag{1}$$

Novel views are rendered using a 6D light-field parametrization $\mathbf{r} = (\mathbf{o}, \mathbf{d})$ of the scene. Each pixel to be rendered is described by the camera position $\mathbf{o}$ and the normalized ray direction $\mathbf{d}$ pointing from the camera through the center of the pixel in the image plane. As shown in Fig. 2 (left), the *Decoder Transformer* $\mathcal{D}$ uses these rays $\mathbf{r}$ as queries to attend into the SLSR, thereby aggregating localized information from the scene, and ultimately produces the RGB color prediction

$$C(\mathbf{r}) = \mathcal{D}_\theta(\mathbf{r} \mid \{\mathbf{z}_j\}). \tag{2}$$

Given a dataset of images $\{\mathbf{I}_{s,i}^{\text{gt}}\}$ from different scenes indexed by $s$, the model is trained end-to-end using an L2 reconstruction loss for novel views:

$$\arg\min_\theta \ \sum_s \mathbb{E}_{\mathbf{r} \sim \mathbf{I}_{s,i}^{\text{gt}}} \|C(\mathbf{r}) - \mathbf{I}_{s,i}^{\text{gt}}(\mathbf{r})\|_2^2. \tag{3}$$

## 2.2 Scene decomposition

SRT's latent representation, the SLSR, has been shown to contain enough information to perform downstream tasks such as semi-supervised semantic segmentation [29]. However, the size of the SLSR is directly determined by the number and resolution of the input images, and there is no clear one-to-one correspondence between the SLSR tokens and objects in the scene.

To obtain an object-centric scene representation, we incorporate the Slot Attention [23] module into our architecture. Slot Attention converts the SLSR $\{\mathbf{z}_j\}$ into the *Slot Scene Representation (SlotSR)*, a set of object *slots* $\{\mathbf{s}_n \in \mathbb{R}^h\}$. Different from the size of the SLSR, the size $N$ of the SlotSR is chosen by the user.

We initialize the set of object slots using learned embeddings $\{\hat{\mathbf{s}}_n \in \mathbb{R}^h\}$. The Slot Attention module then takes the following form for learned linear projections $W_v$, $W_z$, and $W_s$ and a learned update function $\mathcal{U}_\theta$:

$$\mathbf{s}_n = \mathcal{U}_\theta\left(\hat{\mathbf{s}}_n, \frac{\sum_{j=1}^J \mathbf{A}_{n,j} W_v \mathbf{z}_j}{\sum_{j=1}^J \mathbf{A}_{n,j}}\right), \quad \text{with} \quad \mathbf{A}_{n,j} = \frac{\exp\left((W_s \hat{\mathbf{s}}_n)^T W_z \mathbf{z}_j\right)}{\sum_{l=1}^N \exp\left((W_s \hat{\mathbf{s}}_l)^T W_z \mathbf{z}_j\right)}. \tag{4}$$

The attention matrix $\mathbf{A}$ is used to aggregate input tokens using a *weighted mean*. Different from commonly used cross-attention [36], it is normalized over the output axis, *i.e.*, the set of slots, instead of the input tokens. This enforces an exclusive grouping of input tokens into object slots, which serves as an inductive bias for decomposing the SLSR into individual per-object representations. Following Locatello et al. [23], we use a gated update for $\mathcal{U}_\theta$ in the form of a GRU [6] followed by a residual MLP and we apply LayerNorm [2] to both inputs and slots.

## 2.3 Efficient object-centric decoding

To be able to extract arbitrary-view object decompositions from the model, we propose the novel *Slot Mixer (SM)*: a powerful, yet efficient object-centric decoder. The SM module is shown in Fig. 2 (center) and consists of three components: *Allocation Transformer*, *Mixing Block*, and *Render MLP*.

**Allocation Transformer.** The goal of the *Allocation Transformer* is to derive which slots are relevant for the given ray $\mathbf{r} = (\mathbf{o}, \mathbf{d})$. In essence, this module derives object location and boundaries and resolves occlusions. Its architecture is similar to SRT's Decoder Transformer. It is a transformer that uses the target ray $\mathbf{r}$ as the query to repeatedly attend into and aggregate features from the SlotSR:

$$\mathbf{x} = \mathcal{D}_\theta(\mathbf{r} \mid \{\mathbf{s}_n\}) \tag{5}$$

Most compute in the Allocation Transformer is spent on the query, allowing it to scale gracefully to large numbers of objects in the SlotSR. However, unlike in SRT, its output is not used directly for the RGB color estimate.

---

[2]The images may optionally contain pose information, which we omit here for notational clarity.

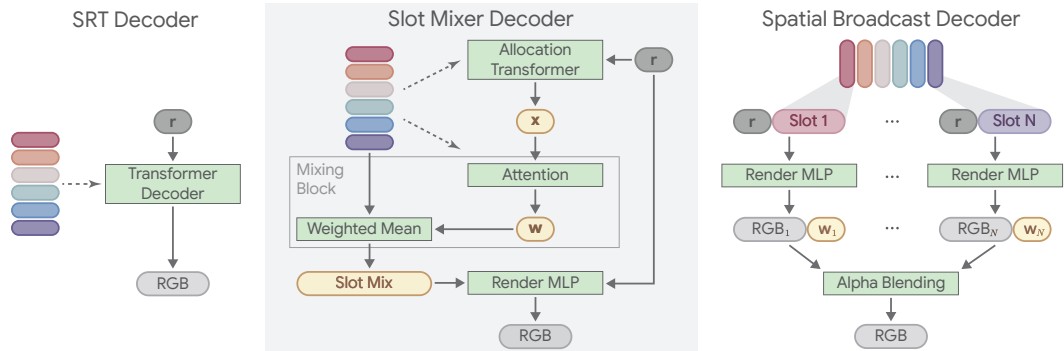

Figure 2: **Decoder architectures** – Comparison between SRT, the novel *Slot Mixer (SM)*, and *Spatial Broadcast (SB)* decoders. The SRT decoder uses an efficient Transformer that scales gracefully to large numbers of objects, but it fails to produce novel-view object decompositions. The commonly used SB model decodes each slot independently, leading to high memory and computational requirements. The proposed SM decoder combines SRT's efficiency with SB's object decomposition capabilities. Details in Secs. 2.3 and 2.4.

**Mixing Block.** Instead, the resulting feature $\mathbf{x}$ is passed to the *Mixing Block*, which computes a normalized dot-product similarity $\mathbf{w}$ with the SlotSR matrix $\mathbf{S} \in \mathbb{R}^{N \times h}$, *i.e.*, the slots in arbitrary, but fixed order. This similarity is then used to compute a weighted mean of the original slots:

$$Q = W_Q \mathbf{x}, \quad K = W_K \mathbf{S}^T, \qquad \mathbf{w} = \text{softmax}(K^T Q), \qquad \bar{\mathbf{s}} = \mathbf{w}^T \mathbf{S}, \qquad (6)$$

where $W_Q$ and $W_K$ are learned linear projections. Notably, the weight $\mathrm{w}_i$ of each slot is scalar, and unlike standard attention layers, no linear maps are computed for the slots, *i.e.* the Allocation Transformer is solely responsible for *mixing* the slots, not for *decoding* them. The slot weights can be seen as novel-view object assignments that are useful for visual inspection of the learned representation and for quantitative evaluation, see Sec. 4.

**Render MLP.** Finally, the *Render MLP* decodes the original query ray $\mathbf{r}$ conditioned on the weighted mean $\bar{\mathbf{s}}$ of the SlotSR into the RGB color prediction:

$$C(\mathbf{r}) = \mathcal{M}_\theta(\mathbf{r} \mid \bar{\mathbf{s}}) \qquad (7)$$

We call the resulting model as shown in Fig. 1 with the SRT encoder, incorporated Slot Attention module and the Slot Mixer decoder *Object Scene Representation Transformer (OSRT)*. All parameters are trained end-to-end using the L2 reconstruction loss as in Eq. (3).

## 2.4 Alternative decoder architecture

Our novel Slot Mixer differs significantly from the standard choice in the literature where *Spatial Broadcast* (SB) decoders [37] (Fig. 2, right) are most commonly used in conjunction with Slot Attention [23, 33, 40]. We present an adaptation thereof to OSRT as an alternative to the SM decoder.

In SB decoders, the query ray is decoded for each slot $\mathbf{s}_n$ independently using a shared MLP $\mathcal{M}$:

$$\mathrm{c}_{n,\mathbf{r}}, \alpha_{n,\mathbf{r}} = \mathcal{M}_\theta(\mathbf{r} \mid \mathbf{s}_n) \qquad (8)$$

For each ray $\mathbf{r}$, this produces color estimates $\mathrm{c}_\mathbf{r} \in \mathbb{R}^N$ and logits $\alpha_\mathbf{r} \in \mathbb{R}^N$ with each value corresponding to a different slot. The final RGB color estimate is then calculated by a normalized, weighted mean:

$$\mathbf{w} = \text{softmax}(\alpha_\mathbf{r}), \qquad C(\mathbf{r}) = \mathbf{w}^T \mathrm{c}_\mathbf{r} \qquad (9)$$

In the SB decoder, the slots therefore *compete* for each pixel through a softmax operation.

We note a major disadvantage of this popular decoder design: it does not scale gracefully to larger numbers of objects or slots, as the full decoder has to be run on each slot. This implies a linear increase in memory and computational requirements *of the entire decoder*, which is often inhibitive,

especially in training. In practice, most pixels are fully explained by a single slot, *i.e.* almost all of the compute of the decoder is spent on resolving the most prominent slot, and the rest go unused.

The proposed Slot Mixer solves this by employing the scalable *Allocation Transformer* for blending between slots more efficiently, while only *a single* weighted mean of all slots must be fully decoded by the *Render MLP*. We further investigate the choice of decoders empirically in Sec. 4.2.

## 3    Related works

**Neural rendering.**    Neural rendering is a large, promising field that investigates the use of machine learning for graphics applications [34]. Recently, NeRF [25] has sparked a renewed wave of interest in this field by optimizing an MLP to parameterize a single volumetric scene representation and demonstrating photo-realistic results on real-world scenes, later also for uncurated in-the-wild data [24]. Further methods based on NeRF are able to generalize across scenes by means of reprojecting 3D points into 2D feature maps [35, 39], though they lack a *global* 3D-based scene representation that could be readily used for downstream applications.

Meanwhile, there have been early successes with global latent models [15, 31], though these rarely scale beyond simple datasets of single oriented objects on uniform backgrounds [20]. Alternative approaches produce higher-quality results by employing a computationally expensive auto-regressive generative mechanisms that does not produce spatially or temporally consistent results [28]. Recently, the Scene Representation Transformer (SRT) [29] has been proposed as a global-latent model that efficiently scales to highly complex scenes by means of replacing the volumetric parametrization with a light field formulation.

**Object-centric learning.**    Prior works on object-centric learning, such as MONet [3], IODINE [9], SPACE [21] and Slot Attention [23] have demonstrated that it is possible to learn models that decompose images into objects using simple unsupervised image reconstruction objectives. These methods typically use a structured latent space and employ dedicated inductive biases in their architectures and image decoders. To handle depth and occlusion in 3D scenes, these methods typically employ a generative model which handles occlusion via alpha-blending [9, 23] or by, e.g., generating objects ordered by their distance to the camera [1, 21]. We refer to the survey by Greff et al. [10] for an overview. Recent progress in this line of research applies these core inductive biases to work on images of more complex scenes [30] or video data [17, 19]. In our OSRT model, we make use of the ubiquitous Slot Attention [23] module because of its efficiency and effectiveness.

**3D object-centric methods.**    Several recent works have extended self-supervised object-centric methods to 3D scenes [5, 33, 40]. One of the first such approaches is ROOTS [5], a probabilistic generative model that represents the scene in terms of a 3D feature map, where each 3D "patch" describes the presence (or absence) of an object. Each discovered object is independently rendered using a GQN [7] decoder and the final image is recomposed using Spatial Transformer Networks [14].

Further prior works that are the most relevant to our method are ObSuRF [33] and uORF [40], both combining learned volumetric representations [25] with Slot Attention [23] and Spatial Broadcast decoders [37]. Both methods have been shown to be capable of modeling 3D scenes of slightly higher complexity than CLEVR [16].

ObSuRF's architecture is based on NeRF-VAE [20]. It bypasses the extraordinary memory requirements of the Spatial Broadcast decoder combined with volumetric rendering during training by using ground-truth depth information, thereby needing only 2 samples per ray instead of hundreds. During inference, at the absence of ground-truth depth information, ObSuRF however suffers from the expected high computational cost of object-centric volumetric rendering.

uORF is based on an encoder-decoder architecture that explicitly handles foreground (FG) and background (BG) separately through a set of built-in inductive biases and assumptions on the dataset. For instance, the dedicated BG slot is parameterized differently, and FG slots are encouraged to only produce density inside a manually specified area of the scene. The model is trained using a combination of perceptual losses, and optionally also with an adversarial loss [8].

In order to obtain the results for ObSuRF and uORF, we used the available official implementations, and consulted the respective authors of both methods to ensure correct adaptation to new datasets.

Finally, a separate line of work considers purely generative object-centric 3D rendering without the ability to render novel views of a specifically provided input scene. GIRAFFE [26] addresses this problem by combining volumetric rendering with a GAN-based [8] loss. It separately parameterizes object appearance and 3D pose for controllable image synthesis. Inspired thereof, INFERNO [4] combines a generative model with a Slot Attention-based inference model to learn object-centric 3D scene representations. We do not explicitly compare to INFERNO as it is similar to ObSuRF and uORF, while only shown capable of modeling CLEVR-like scenes of lower visual complexity.

## 4 Experiments

To investigate OSRT's capabilities, we evaluate it on a range of datasets. After confirming that the proposed method outperforms existing methods on their comparably simple datasets, we move on to a more realistic, highly challenging dataset for all further investigations. We evaluate models by their novel view reconstruction quality and unsupervised scene decomposition capabilities qualitatively and quantitatively. We further investigate OSRT's computational requirements compared to the baselines and close this section with some further analysis into which ingredients are crucial to enable OSRT's unsupervised scene decomposition qualities in challenging settings.

**Setting and evaluation metrics.** The models are trained in a novel view synthesis (NVS) setup: on a dataset of *scenes*, we train the models to produce novel views of the same scene parameterized by target camera poses. For evaluation, we run the models on a held-out set of test scenes and render multiple novel views per scene.

As quantitative metrics, we report PSNR for pixel-accurate reconstruction quality and adopt the standard foreground Adjusted Rand Index (FG-ARI) [13, 27] to measure object decomposition. Crucially, we compute FG-ARI on all rendered views together, such that object instances must be consistent between different views. This makes our metric sensitive to 3D-inconsistencies that may especially plague light field models which do not explicitly enforce this in contrast to volumetric methods. We analyze and discuss this choice further in Sec. 4.2.

For qualitative inspection of the inferred scene decompositions, we visualize for each rendered pixel the slot with the highest weight $w_i$ and color-code the slots accordingly.

**Datasets.** We run experiments on several datasets in increasing order of complexity.

**CLEVR-3D [33].** This is a recently proposed multi-camera variant of the CLEVR dataset, which is popular for evaluating object decomposition due to its simple structure and unambiguous objects. Each scene consists of 3–6 basic geometric shapes of 2 sizes and 8 colors randomly positioned on a gray background. The dataset has ∼35k training and 100 test scenes, each with 3 fixed views: the two target views are the default CLEVR input view rotated by $120°$ and $240°$, respectively.

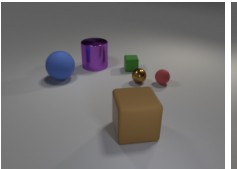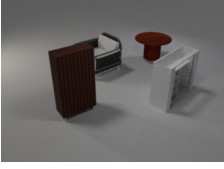

Figure 3: Example views of scenes from CLEVR-3D (left) and MSN-Easy (right).

**MultiShapeNet-Easy (MSN-Easy) [33].** This dataset is similar in structure to CLEVR-3D, however, 2–4 upright ShapeNet objects sampled from the *chair*, *table* and *cabinet* classes (for a total of ∼12k objects) now replace the geometric solids. The dataset has 70k training and 100 test scenes.

**MultiShapeNet-Hard (MSN-Hard) [29].** MSN-Hard has been proposed as a highly challenging dataset for novel view synthesis. In each scene, 16-32 ShapeNet objects are scattered in random orientations. Realistic backgrounds and HDR environment maps are sampled from a total set of 382 assets. The cameras are randomly scattered on a half-sphere around the scene with varying distance to the objects. It is a highly demanding dataset due to its use of photo-realistic ray tracing [11], complex arrangements of tightly-packed objects of varying size, challenging backgrounds, and nontrivial camera poses including almost horizontal views of the scene. The dataset has 1M training scenes, each with 10 views, and we use a test set of 1000 scenes. The ∼51k unique ShapeNet objects are taken from *all* classes, and they are separated into a train and test split, such that the test set not only contains novel arrangements, but also novel objects. We re-generated the dataset as the original provided by Sajjadi et al. [29] does not include ground-truth instance labels necessary for our quantitative evaluation. We will make the dataset publicly available.

Table 1: **Quantitative results** – OSRT outperforms ObSuRF across all metrics and datasets with the exception of CLEVR-3D, where FG-ARI is nearly identical. On MSN-Hard, OSRT is able to encode multiple input views to further improve object decomposition and reconstruction performance.

| | CLEVR-3D [33] | | MSN-Easy [33] | | MSN-Hard [29] | | |
| | ObSuRF | OSRT (1) | ObSuRF | OSRT (1) | ObSuRF | OSRT (1) | OSRT (5) |
|---|---|---|---|---|---|---|---|
| PSNR | 33.69 | **39.98** | 27.41 | **29.74** | 16.50 | 20.52 | **23.54** |
| FG-ARI | **0.978** | 0.976 | 0.940 | **0.954** | 0.280 | 0.619 | **0.812** |

## 4.1 Comparison with prior work

Tab. 1 shows a comparison between OSRT and the strong ObSuRF [33] baseline. On the simpler datasets CLEVR-3D and MSN-Easy, ObSuRF produces reasonable reconstructions with accurate decomposition, though OSRT achieves significantly higher PSNR and similar FG-ARI.

On the more realistic MSN-Hard, ObSuRF achieves only low PSNR and FG-ARI. OSRT on the other hand still performs solidly on both metrics. Furthermore, while ObSuRF can only be conditioned on a single image, OSRT is optionally able to ingest several. We report numbers for OSRT (5), our model with 5 input views, on the same dataset and see that it substantially improves both reconstruction and decomposition quality.

At the same time, OSRT renders novel views at $32.5$ fps (frames per second), more than $3000\times$ faster than ObSuRF which only achieves $0.01$ fps, both measured on an Nvidia V100 GPU. This speedup is the result of two multiplicative factors: OSRT's light field formulation is $\sim100\times$ faster than volumetric rendering and the novel Slot Mixer is $\sim30\times$ faster than the SB decoder here. Finally, OSRT does not need ground-truth depth information during training.

Fig. 4 shows qualitative results for the realistic MSN-Hard dataset. It is evident that ObSuRF has reached its limits, producing blurry images with suboptimal scene decompositions. OSRT on the other hand still performs solidly, producing much higher-quality images and decomposing scenes reasonably well from a single input image. With more images, renders become sharper and decompositions more accurate.

As the second relevant prior method, we have performed experiments with uORF [40]. Despite guidance from the authors, uORF failed to scale meaningfully to the most interesting MSN-Hard dataset. This mainly resulted from a lack of model capacity. Additionally, the large number of objects in this setting led to prohibitive memory requirements of the method, forcing us to lower model capacity even further, or to run the model with fewer slots to be able to fit it even on an Nvidia A100 GPU with 40 GB of VRAM. We describe this further with results in the appendix.

## 4.2 Ablations and model analysis

We conduct several studies into the behavior of the proposed model including changes to architecture, training setup, or data distribution. Unless stated otherwise, all investigations in this section are conducted on the MSN-Hard dataset with 5 input views. Further qualitative results for these experiments are provided in the appendix.

**Decoder architecture.** As described in Sec. 2.4 and shown in Fig. 2, the novel Slot Mixer decoder differs significantly from the default SRT decoder [29], as well as from the Spatial Broadcast (SB) decoder that is commonly used in conjunction with Slot Attention. To show the strengths of the SM decoder, we compare it with these alternative decoders switched in to the model, see Tab. 2.

Table 2: **OSRT decoder variants** – Slot Mixer combines SRT's efficiency with SB's object decomposition abilities. Results on simpler datasets are shown in Tab. 4.

| MSN-Hard | PSNR | FG-ARI | FPS |
|---|---|---|---|
| SRT Decoder | **24.40** | 0.330 | **40.98** |
| Spatial Broadcast (SB) | 23.35 | 0.801 | 1.39 |
| Slot Mixer (SM) | 23.54 | **0.812** | 32.47 |

| Input | ObSuRF | OSRT (1) | OSRT (5) | Target |
|-------|--------|----------|----------|--------|

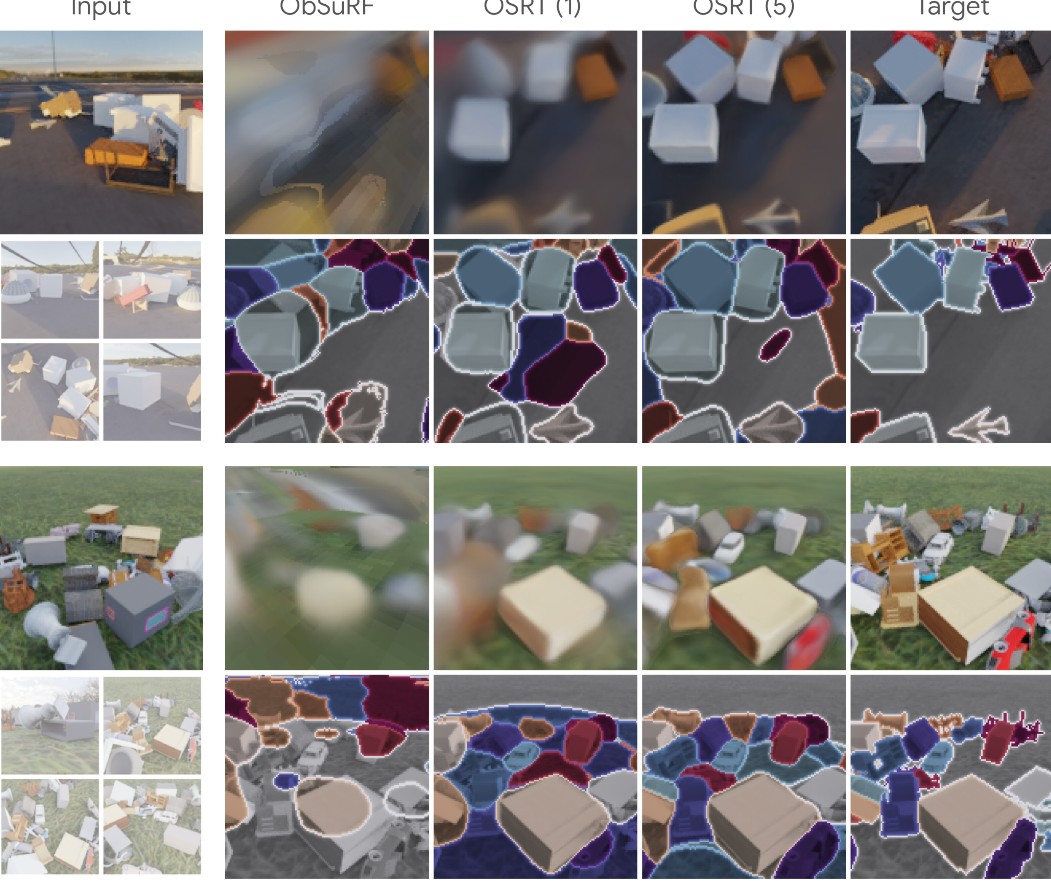

Figure 4: **Qualitative results on MSN-Hard** – Comparison of our method with one and five input views with ObSuRF, which can only operate on a single image. OSRT produces sharper renders with better scene decompositions.

While the SRT decoder achieves slightly higher reconstruction quality due to the powerful transformer, it does not yield useful object decompositions as a result of the global information aggregation across all slots, disincentivizing object separation in the slots. It is the key design choice in Slot Mixer to only use a transformer for slot mixing, but not for decoding the slots, that leads to good decomposition.

The SB decoder performs similarly to Slot Mixer both in terms of reconstruction and decomposition. However, due to the slot-wise decoding, it requires considerably more memory, which can often be prohibitive during training, and hamper scalability to more complex datasets or large numbers of objects. For the same reason, it also requires significantly more compute at training and for inference.

**Role of novel view synthesis.** Our experiments have demonstrated OSRT's strengths in the default novel view synthesis (NVS) setup. We now investigate the role of NVS on scene decomposition, on the complex MSN-Hard dataset.

To this end, we surgically remove the NVS component from the method, while keeping all other factors unchanged, by training OSRT with the input views equaling the target views to be reconstructed. This effectively turns OSRT into a multi-2D image auto-encoder, albeit with 3D-centric poses rather than pure 2D positional encoding for ray parametrization.

The resulting method achieves a much better PSNR of $28.14$, $4.60$ db higher than OSRT in the NVS setup ($23.54$). This is expected, as the model only needs to *reconstruct* the target images rather than needing to generate *arbitrary novel views* of the scene. However, the model fails to decompose the scene into objects, only achieving an FG-ARI of $0.198$ compared to OSRT's FG-ARI of $0.812$, demonstrating the advantage of using NVS as an auxiliary task for unsupervised scene decomposition.

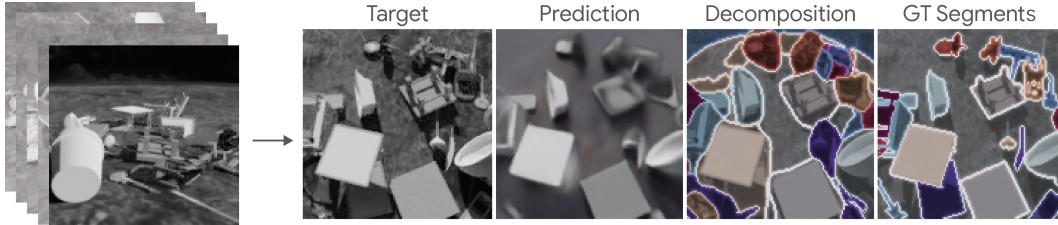

Figure 5: OSRT trained on MSN-Hard (in color) is evaluated with grayscale input images at test time. We observe that the model generalizes remarkably well to this out-of-distribution setting. This indicates that OSRT does not just rely on color for decomposition.

**Robustness.** We begin with a closer glance at the results obtained by OSRT (5) on MSN-Hard based on the number of objects in the scene. We find that the FG-ARI scores for scenes with the smallest (16) and largest (31) number of objects are within an acceptable range: $0.854$ *vs.* $0.753$.

To explore whether OSRT mainly relies on RGB color for scene decomposition, we evaluate it on a grayscale version of the difficult test set of MSN-Hard. Note that the model was trained on RGB and has not encountered grayscale images at all during training. We find that OSRT generalizes remarkably well to this out-of-distribution setting, still producing satisfactory reconstructions and scene decompositions that are almost up to par with the colored test set at FG-ARI of $0.780$ *vs.* $0.812$ on the default RGB images. Fig. 5 shows an example result for this experiment.

We also consider to what extent OSRT is capable of leveraging additional context at test time in the form of additional input views. We train OSRT (3) with three input views and then evaluate it using three (PSNR: $22.75$, FG-ARI: $0.794$) and five input views (PSNR: $23.47$, FG-ARI: $0.813$). These results clearly indicate that additional input views can be used to boost performance at test-time.

Finally, we train OSRT in two setups inspired by SRT variants [29]: UpOSRT, trained without input view poses, and VOSRT, using volumetric rendering. We find that both of these achieve good reconstruction quality at PSNR's of $22.42$ and $21.38$ and meaningful scene decompositions at $0.798$ and $0.767$ FG-ARI, respectively.

**Scene editing.** We investigate OSRT's ability for simple scene editing. Fig. 6 (left) shows the a novel view rendered by OSRT. When the slot corresponding to the large object is removed from the SlotSR (center), the rendered image reveals previously occluded objects. We can go one step further by *adding* a slot from a different scene (right) to the SlotSR, leading to the object being rendered in place with correct occlusions.

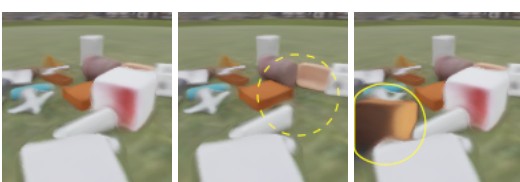

Figure 6: Novel view (left) with a slot removed (center) or a slot added from another scene (right).

**3D consistency.** An important question with regards to novel view synthesis is whether the resulting scene and object decomposition are 3D-consistent. Prior work has demonstrated SRT's spatial consistency despite its light field formulation [29]. Here, we investigate OSRT 3D-consistency with regards to the learned decomposition.

To measure this quantitatively, we compute the ratio between the FG-ARI as reported previously—computed on all views together—and the 2D-FG-ARI which is the average of the FG-ARI scores for each individual target view. While FG-ARI takes into account 3D consistency, 2D-FG-ARI is unaffected by 3D inconsistencies such as permuting slot-assignments between views. By definition, 2D-FG-ARI is an upper bound on FG-ARI. Hence, an FG-ARI ratio of 1.0 indicates that the novel views are perfectly 3D-consistent, while lower values indicate that some inconsistency took place.

We observe that our approach consistently achieves very high 3D consistency in scene decompositions, with an FG-ARI ratio of $0.940$ on the challenging MSN-Hard dataset. With 5 input views, OSRT achieves an even higher ratio of $0.987$. Despite its volumetric parametrization, we find that ObSuRF often fails to produce consistent assignments with an FG-ARI ratio of only $0.707$.

Table 3: **OSRT generalization on CLEVR-3D** – OSRT trained on scenes with 3-6 objects is tested on scenes with 3–6, and 7-10 objects, respectively. Both with learned and random slot initializations, OSRT generalizes reasonably to the presented out-of distribution (OOD) setting with more objects than during training. The slight drop in performance is mostly a result of more pixels being covered by objects rather than the simpler background.

| 3-6 objects (IID) | PSNR | FG-ARI | | 7-10 objects (OOD) | PSNR | FG-ARI |
|---|---|---|---|---|---|---|
| Learned init. | 40.84 | 0.996 | | Learned init. | 33.03 | 0.955 |
| Random init. | 38.14 | 0.988 | | Random init. | 33.36 | 0.968 |

**Out-of-distribution generalization.** We investigate the ability of the model to generalize to more objects at test time than were observed at training time. To this end, we trained OSRT either with 7 randomly initialized slots, or with 11 slots using a learned initialization, on CLEVR-3D scenes containing up to 6 objects. At test time, we evaluate on scenes containing 7-10 objects by using 11 slots for both model variants.

The results are shown in Tab. 3. We find that generalization performance in terms of PSNR and FG-ARI is similarly good for both models, with a slight advantage for the model variant with random slot initialization. In terms of in-distribution performance, learned initialization appears to have an advantage in both metrics.

### 4.3 Limitations

In our experimental evaluation of OSRT, we came across the following two limitations that are worth highlighting: 1) while the object segmentation masks produced by OSRT often tightly enclose the underlying object, this is not always the case and we find that emergent masks can "bleed" into the background, and 2) OSRT can, for some architectural choices, fall into a failure mode in which it produces a 3D-spatial Voronoi tessellation of the scene instead of clear object segmentation, resulting in a substantial drop in FG-ARI.

While the alternative, yet much more inefficient, SB decoder does not seem to be affected by this, mask bleeding [9] effects and tesselation failure modes [18] are not uncommon in object-centric models. We show more examples and comparisons between the different architectures in the appendix.

## 5   Conclusion

We present *Object Scene Representation Transformer (OSRT)*, an efficient and scalable architecture for unsupervised neural scene decomposition and rendering. By leveraging recent advances in object-centric and neural scene representation learning, OSRT enables decomposition of complex visual scenes, far beyond what existing methods are able to address. Moreover, its novel *Slot Mixer* decoder enables highly efficient novel view synthesis, making OSRT more than $3000\times$ faster than prior works. Future work has the potential to elevate such methods beyond modeling static scenes, instead allowing for moving objects. We believe that our contributions will significantly support model design and scaling efforts for object-centric geometric scene understanding.

## Acknowledgments

We thank Karl Stelzner and Hong-Xing Yu for their responsive feedback and guidance on the respective baselines, Mike Mozer for helpful feedback on the manuscript, Noha Radwan and Etienne Pot for help and guidance with datasets, and the anonymous reviewers for the helpful feedback.

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
