# OpenReview forum: "Object Scene Representation Transformer"
_NeurIPS.cc/2022/Conference — NeurIPS 2022 Accept_

### Official Review · Reviewer_HAVj · 2022-07-06

**Rating:** 5
**Confidence:** 4
**Soundness:** 3 good
**Presentation:** 3 good
**Contribution:** 2 fair

**Summary:**

The paper presents a method to infer an object-centric scene representation aware of 3D geometry from multiple views of a scene. Similar to other works, it uses slot attention to segment scenes into slots, and it uses a ray-casting approach to rendering 2D views of the represented scene. However, the authors propose a novel decoder, based on Scene Representation Transformers (SRT), that only requires a single evaluation per ray query regardless of the number of objects in the representation. This makes the model faster than previous alternatives, while also achieving good quantitative metrics on different datasets.

**Questions:**

- How is the background handled?
- Differently from Slot Attention, it seems that the initial slot representation is learned. How does that affect the model performance and how does it affect its generalization capabilities to an a different number of objects from training/testing?
- How does the model compare to UORF quantitatively on the simpler datasets proposed in that paper?
- The segmentations compared on the FG-ARI metric do not take into account that some object segmentations might be covering the background. How do the metrics change when using the ARI metric

**Limitations:**

The authors mention limitations of their work and ethical considerations. The limitations of their work could be better analyzed by conducting some of the experiments suggested in the Questions section.

**Strengths And Weaknesses:**

**Strengths:**

+ Clear presentation:

The problem, method and results are presented clearly and it is easy to follow the paper.

+ Fast approach to decoding 3D object-centric representations:

The model has much lower computational costs per iteration given that the number of evaluations per ray do not scale linearly with the number of objects. This makes the model fast and more scalable than other alternatives.

+ Good performance compared to ObjSURF:

The model outperforms ObjSURF both quantitatively and qualitatively on different datasets.


**Weaknesses:**

- Missing references:

The paper is missing numerous relevant references. Specifically, it is missing the whole body of work of Sungjin Ahn's team, with ROOTS[1] being directly related to this paper, and relevant work by others [2,3].

- Evaluation could be improved:

While the paper compares in three datasets of increasing difficulty, the comparison is mostly to ObjSURF. The paper only compares to UORF on the harder dataset, in which it does not work well due to the high number of objects and computational costs associated. While this is a good comparison to highlight the better scaling capabilities of the proposed model (and this is an important point), it does not resolve the issue of how this model compares to competing approaches when they can actually compete. An easy way to solve this would be to train UORF and other previous methods on the simpler datasets, or compare on one of the proposed datasets in the UORF paper. In fact, there are experiments with different backgrounds and more complex objects in that paper, that are similar to the proposed datasets in this paper.
It would also be interesting to understand the quality of the segmentations by reporting regular ARI metrics, instead of just FG-ARI metrics, as these can be misleading when objects are not segmented tightly.


**Detailed comments:**

Overall this is a promising submission. While the idea of having an object-centric SRT model is relatively straightforward, the proposed decoder is novel and has clear advantages in terms of computational costs while still being interpretable. However, the paper could be improved with a more thorough literature review and a better comparison to previous approaches based on this literature review. There are a few questions to be answered, but overall I believe this can be a good submission with some modifications. I am currently arguing for its acceptance, and I would be happy to increase my score if the authors address my comments.

**References:**
[1] Chen, Chang, Fei Deng, and Sungjin Ahn. "ROOTS: Object-Centric Representation and Rendering of 3D Scenes." J. Mach. Learn. Res. 22 (2021): 259-1.
[2] Castrejon, Lluis, Nicolas Ballas, and Aaron Courville. "INFERNO: Inferring Object-Centric 3D Scene Representations without Supervision." (2021).
[3] Anciukevicius, Titas, Christoph H. Lampert, and Paul Henderson. "Object-centric image generation with factored depths, locations, and appearances." arXiv preprint arXiv:2004.00642 (2020).

---

> ### Author Response · Authors · 2022-08-02
> **Response to HAVj**
>
> We thank the reviewer for the positive feedback.
>
> **Related work on 3D object discovery**
>
> We thank the reviewer for the references to additional related works. ROOTS [1*] and INFERNO [2*] are indeed closely related in that they address a similar problem, yet target significantly simpler experimental settings and datasets compared to our work. As such, we find that uORF [34] and ObSuRF [27] serve as better representatives for our setting: only they have been demonstrated on data of significantly higher visual complexity than CLEVR.
>
> We will add these references and further update our literature review to include OCIG [3*] and other related methods from Sungjin Ahn’s lab.
>
>
> **Comparison to uORF and dataset complexity**
>
> We would like to underline that the positive result of uORF on the visually more complex dataset in their paper is limited to using only a single class of objects (chairs). In this setting, the authors achieve strong results when limiting the data to a single model of a simple chair rendered in various colors, while segmentation quality already drops significantly when moving to multiple different chair shapes (see Tab. 2 in the uORF paper).
>
> In terms of dataset complexity, we disagree with the statement that the MSN-H dataset is similar to datasets investigated in uORF. MSN-H is of substantially higher complexity even without considering the number of objects per scene: it contains diverse objects sampled from all ShapeNet classes (~51k unique objects, only novel objects in the test set) in random poses, 382 different realistic HD outdoor backgrounds (as opposed to 50 floor textures in uORF’s most complex dataset), and further contains significantly more challenging camera angles. We believe that this makes MSN-H a more suitable choice for studying the capabilities and limits of this class of models. To study model behavior in datasets with fewer objects per scene, we chose two representative datasets with varying levels of object complexity: CLEVR-3D and MSN-E.
>
> **Background handling in OSRT**
>
> Similar to most methods utilizing Slot Attention (including the original paper [17] and ObSuRF [27]), we do not handle backgrounds in a special way: any slot can bind to (parts of) the background, and we typically find that it is captured by slots that do not bind to objects otherwise. Hence in practice, we choose to have at least 1 slot more than the maximum number of objects that we would like to model in a scene.
>
> uORF [34] assigns a specific slot to handle the background, but their approach requires additional supervision (e.g. in the form of the bounding box of the scene in which objects can appear). We found this solution to be unsatisfactory as it makes strong assumptions about the scene layout, but we agree that a general form of explicit background handling in slot-based models is an interesting avenue for future work.
>
>
> **Segmentation evaluation**
>
> As the background itself can have compositional structure, we find that the full ARI score (i.e. ARI that is evaluated on all pixels including background pixels) is not an ideal metric for evaluation of object-centric models in our settings, as it strongly discounts discovered solutions that separate the background into meaningful parts. In line with prior work, we therefore choose FG-ARI as our main metric for evaluating unsupervised scene decomposition into objects. We will add (non-FG) ARI results to the paper if space permits, and otherwise to the appendix.
>
>
> **Learned vs. random slot initialization**
>
> The question whether using learned (as opposed to random) slot initializations affects generalization to a different number of objects at test time in OSRT is indeed very interesting. Please note that the original Slot Attention paper already proposed both settings of using random vs. learned slot initializations, i.e. this is not one of our novel contributions. To analyze this in OSRT, we perform the following experiment: we trained OSRT either with 7 randomly initialized slots, or with 11 slots using a learned initialization, on CLEVR-3D scenes containing up to 6 objects.
> At test time, we evaluate on scenes containing 7-10 objects by using 11 slots for both model variants. We find that in this setting, generalization performance in terms of PSNR and FG-ARI is similar for both models with a slight advantage for the model variant with random slot initialization. In terms of in-distribution performance, learned initialization appears to have an advantage in both metrics.
>
>
> |3-6 objects (iid)|PSNR|FG-ARI|
> |-------------------|-------|--------|
> |Learned init.|40.84|0.996|
> |Random init.|38.14|0.988|
>
>
> |7-10 objects (ood)|PSNR|FG-ARI|
> |--------------------|-------|--------|
> |Learned init.|33.03|0.955|
> |Random init.|33.36|0.968|
>
> We generally found that learned slot initializations resulted in more stable training, especially on the more complex datasets, hence we opted for this as our default setting.

---

### Official Review · Reviewer_3uqV · 2022-07-11

**Rating:** 7
**Confidence:** 4
**Soundness:** 3 good
**Presentation:** 3 good
**Contribution:** 3 good

**Summary:**

- The paper addresses the problem of unsupervised decomposition of scenes into a discrete set of objects. The idea is that novel view synthesis requires reasoning about the geometry which should also be helpful for discovering objects without additional supervision. A novel transformer-based approach for object centric 3D representation learning is introduced which aims to overcome limitations of previous methods regarding generalization to complex scenes due to high computational costs.
- The method uses a slot-attention mechanism in combination with a light-field rendering formulation using a transformer conditioned on a viewpoint-dependent latent representation derived from these slots. To avoid high computational costs, an early fusion scheme is presented that mixes slots early and thus has to run the decoder only once and not per slot.
- Compared to [27], the proposed approach does not require depth supervision for efficiency as it relies on an SRT [23] based decoder for efficient rendering instead of volumetric rendering approaches.

**Questions:**

- Fig. 3 in the supplementary seems to indicate that the Spatial Broadcast (SB) decoder suffers less from oversegmenting. Qualitatively it also seems to be more accurate around object boundaries but the quantitative results in Tab. 3 say the opposite. What could explain this discrepancy and could the proposed Slot Mixer (SM) decoder benefit from an explicit background modeling to avoid these problems?

**Limitations:**

Limitations and potential negative societal impact have been addressed adequately.

**Strengths And Weaknesses:**

- Strengths
    - The introduction provides a good motivation towards the goal of learning about compositions and the geometry of scenes solely from observations. The related works section provides good context regarding neural rendering, object centric learning and specific differences to closest related works on object centric 3d learning.
    - Moving unsupervised 3d scene decomposition towards more complex and realistic scenes is an important goal with many applications. The proposed method demonstrates good performance on the more complex MSN-Hard dataset and could serve as a valuable baseline, especially in combination with the dataset's ground-truth instance labels.
    - The proposed approach is simple and reasonable. Its evaluation supports the stated claims regarding the presented model.
    - The experimental comparison to [27] (as well as the description of the attempts to train [5] in the supplementary) convincingly demonstrate the advantages of the proposed model in terms of accuracy and runtime, especially when going to more complex datasets.
    - The design of the ablations allows for a good analysis of the different components involved in the approach. The decoder analysis clearly demonstrates how the SM decoder improves decomposition over an SRT decoder [23] and runtime over an SB decoder. The experiments on the role of novel view synthesis for decomposition validate the hypothesis that novel view synthesis indeed helps object decomposition. Evaluating whether the approach is mainly segmenting by color cues is a good test for generalization capabilities of the approach. Scene editing experiments give a nice qualitative indication that slots indeed represent meaningful objects.
- Weaknesses
    - The novelty of the proposed method is limited, as it is a rather straightforward adaptation of slot attention [17] to SRT [23] with an early fusion scheme for faster decoding.
    - Qualitatively (Fig. 3 of the supplementary), the SB decoder still looks better than the proposed SM decoder, although this comes at a large computational cost. The quantitative results suggest that the SM decoder performs better, which makes me question the metric a bit.

---

> ### Author Response · Authors · 2022-08-02
> **Response to 3uqV**
>
> We thank the reviewer for the positive feedback.
>
> **Qualitative vs. quantitative results for Slot Mixer and Spatial Broadcast decoders**
>
> We understand the reviewer's concerns about the apparent mismatch between qualitative and quantitative results. When visualizing unsupervised scene decomposition results qualitatively, a balance has to be struck between accurately visualizing the quantity that the ARI metric evaluates, and more intuitive visualizations that can appear less misleading. In the draft, we have chosen to visualize hard masks, i.e. for each pixel, we identify the **single** most relevant slot and color it accordingly. Note that the ARI metric follows the same scheme by taking an argmax for evaluation, therefore assuming that each pixel can only belong to a single slot.
>
> As to the apparent mismatch between qualitative results and the FG-ARI metric, we would like to emphasize that this commonly used metric only evaluates foreground objects, as background segmentation is ambiguous. Visualizing only the evaluated areas of the images by masking out the background areas leads to the following visualizations that are more in line with the quantity that FG-ARI measures ("Masked Hard Seg." column):
>
> https://anopic.us/mM2Chkmpjifvc7TfYy70WkAxIw7K0vtUQXcis7qd.png
>
> https://anopic.us/92Buxt3gjXceYGHLHJEAFVLwN1YH1Y3ALmzfsNEj.png
>
> https://anopic.us/4aGAXPKkHfCzmSAcxBEGyPokVMfzVO85bNRRwWzT.png
>
> We observe that the Spatial Broadcast decoder makes some mistakes along the object boundaries, leading to very slightly lower FG-ARI on this dataset.
>
> Further, due to always using 32 slots for all MSN-H scenes, multiple slots are free to capture the background in scenes with fewer objects. In practice, we find that this results in background pixels being shared “softly” by multiple slots – a detail that is lost in our argmax-based visualization. In addition to the hard masks as in the paper ("Paper Figure" column), here we also visualize soft masks ("Soft Segmentation" column) by weighting the color of each slot according to its weight for each pixel. This reveals that the background is not necessarily oversegmented, but rather fully modeled by the union of multiple slots.

---

> > ### Comment · Reviewer_3uqV · 2022-08-08
> > **Thank you**
> >
> > Thank you for the comments and visualizations, they explain the mismatch! I feel like FG-ARI might introduce some bias into the evaluation by explicitly ignoring background errors. But given its common use, it makes sense to report it. Even with slightly lower quality of the Slot-Mixer for some object boundaries (e.g. second example of the first linked image), the improved efficiency still makes the proposed approach valuable. Hence, I keep my rating of 7 after reading the other reviews and comments.

---

### Official Review · Reviewer_VX3v · 2022-07-12

**Rating:** 6
**Confidence:** 4
**Soundness:** 3 good
**Presentation:** 3 good
**Contribution:** 3 good

**Summary:**

This work tackles the task of view-synthesis and extends a prior work “Scene Representation Transformer” by introducing a slot-based bottleneck representation. This allows the model to learn disentangled representations using only the standard view-synthesis objectives, and the learned disentanglement shown to be more accurate the prior unsupervised methods. Moreover, this representation also enables object removal (although not more general editing e.g. translation/scaling that prior NeRF based unsupervised disentanglement methods would allow).

The proposed approach is evaluated using the CLEVR and Multi-ShapeNet. (MSN) datasets where it outperforms a prior work ObSURF in both, view synthesis and scene decomposition accuracy. However, the bottleneck representation does limit the model expressivity and the learned model is less accurate at view-synthesis than the base SRT model.

**Questions:**

N/A

**Limitations:**

Yes

**Strengths And Weaknesses:**

Strengths:
+ The overall approach is simple and intuitive. The key idea of adding slots after the SRT encoder makes sense and it would be valuable for the community to see the results.

+ The ablations performed regarding the decoder architecture choice are informative. In particular, the proposed ‘slot mixer’ solution seems like a good compromise in retaining slot properties while being more efficient than the ‘broadcast’ approach in prior work.

+ The reported improvements ObSuRF are clearly significant and highlight the benefits of adding slots in the SRT network i.e. ability to handle variable views and more accurate results.

+ The ablation highlighting the role of view synthesis for learning accurate decomposition was very interesting, and shows necessity of using novel views for learning as opposed to mere auto-encoding objectives.


Weaknesses:
- While the ‘slot mixer’ vs ‘broadcast’ decoder is a useful contribution, the overall approach rather straight-forwardly combines ideas from prior works like UORF and ObSuRF with the SRT approach, and I feel the technical contribution here is a bit limited. That said, I still think this maybe a valuable (if obvious) combination for the community to see.

- As a downside of using the proposed slot representation, the quality of the synthesized views does suffer. While this is only highlighted in Table 2 on one dataset, it would be helpful. to see similar results across all datasets.

- (minor point) I feel the claim of “3000x faster” rendering is a bit misleading as this mostly comes from the speedup obtained by SRT over volume-rendering based methods, and isn’t really a contribution of this work (the ‘slot mixer’ vs ‘broadcast’ speedup could be attributed to this work, but that is a much smaller factor).
——

Overall, I think this is simple but obvious paper. On the one hand, I am not sure this work makes any remarkable technical contributions or empirical findings. But on the other hand, it represents a well executed and sensible combination which the community would benefit from seeing, and I would therefore lean towards accepting it.

---

> ### Author Response · Authors · 2022-08-02
> **Response to VX3v**
>
> We thank the reviewer for the positive feedback.
>
> **Reconstruction quality & Tab. 2 results on simpler datasets**
>
> We thank the reviewer for the recommendation. We ran the suggested experiment and found the following results (MSN-H results copied from Tab. 2 for convenience):
>
> | CLEVR-3D    | PSNR  | FG-ARI |
> |-------------|-------|--------|
> | Slot Mixer  | 39.98 | 0.976  |
> | SRT Decoder | 40.78 | 0.983  |
>
> | MSN-Easy       | PSNR  | FG-ARI |
> |-------------|-------|--------|
> | Slot Mixer  | 29.74 | 0.954  |
> | SRT Decoder | 29.36 | 0.887  |
>
> | MSN-Hard       | PSNR  | FG-ARI |
> |-------------|-------|--------|
> | Slot Mixer  | 23.54 | 0.812  |
> | SRT Decoder | 24.40 | 0.330  |
>
> We did not find significant differences in reconstruction quality between the decoder variants on the simpler datasets, mostly as the quality is already very high with the Slot Mixer decoder. Interestingly, the SRT decoder achieves similar FG-ARI on CLEVR-3D and still reasonable FG-ARI on MSN-E, while it failed to decompose the scene meaningfully on the more realistic MSN-H dataset.
>
> **Speedup compared to baselines**
>
> We agree that the >3000x speedup compared to the baseline is a result of both SRT and the novel Slot Mixer. As reported in Appendix Tab. 2, the real-world measured speedup of the Slot Mixer (32.47 fps) vs. Spatial Broadcast (1.39 fps) decoder is 23.4x which is a substantial difference in practice. Additionally, peak memory consumption is reduced by a similar factor which has a large impact on feasibility especially during model training.
>
> We will improve the relevant sections in the manuscript to clarify how the speedup is a multiplicative result of these two largely independent factors. That said, we believe that one of our major empirical findings lies in demonstrating that the novel view synthesis task is crucial for better scene decomposition, not explicitly 3D volumetric rendering as in prior works.

---

> > ### Comment · Reviewer_VX3v · 2022-08-08
> > **Suggestions for the Final Version**
> >
> > Thanks for the response. I do hope that the final text incorporates the additional explanations/results:
> > a) Includes the extended version of Tab2 reported above instead of only the MSN-H dataset
> > b) has a more qualified statement on the speedup
> >
> > Overall, I would like to keep my current rating as I believe this paper presents a simple and sensible approach which the community should see.

---

### Official Review · Reviewer_jvub · 2022-07-12

**Rating:** 5
**Confidence:** 2
**Soundness:** 3 good
**Presentation:** 4 excellent
**Contribution:** 3 good

**Summary:**

The paper proposes Object Scene Representation Transformer, a framework for learning scene representation. The framework combines scene representation with the Slot Attention that constructs a bottleneck such that object-centric representation naturally emerges from learning. The proposed method is fast and achieves good performance.

**Questions:**

See above.

**Strengths And Weaknesses:**

Let me start by honestly stating that my research background is in representation learning not in 3D understanding or rendering, therefore my comments should not serve as the key evidence for deciding the outcome of this submission.

The paper is well written and easy to understand even for someone who doesn’t work in the field. The technical details are sound and the claims are fair. The experimental evaluation seems comprehensive. The performance looks good.

As my background is in representation learning, I’d like to raise a few questions on its application in real-world and complex scenes. Object centric representation is mostly learned through adding inductive biases into a system, such as the Slot Attention adopted in the paper. A bottleneck stage can force a neural network to group pixels into a compact representation, and object *naturally emerges*. However, in my opinion and from my research experience, nothing naturally emerges in a neural network – it does what it does because we, the creators, told it so. In other words, we achieve this grouping by injecting inductive biases into the architecture.

I have always been cautious about injecting inductive biases because doing so mostly helps *low data, low compute* regime, but many times harms a model’s capacity to scale to more data and more compute. Take the example of ViTs vs. CNNs – ViT models have much less inductive bias but they need more data to train and start outperforming CNNs (by a significant margin) as the model sizes grow. Inductive bias is a double-edged sword and any practice of adding inductive bias should be well motivated and examined with extreme caution.

Back to the scope of this submission – I found that the datasets used here are rendered rather than from real-world. Naturally in these datasets object properties, such as numbers, lighting, materials, are randomized in a limited range. This setting seems to match the concerning low data, low compute criteria, therefore more inductive biases, especially dataset engineered ones, will unsurprisingly shine among baselines. It may not be the case for real-world data in which case the scalability and capacity of a model can truly be tested.

I’d like to note the above comments are made without a comprehensive investigation of common practice in similar papers and domains. I plan to take all reviewers’ comments into consideration, and after reviewers’ discussion a further recommendation will be made.

---

> ### Author Response · Authors · 2022-08-02
> **Response to jvub**
>
> We thank the reviewer for the positive feedback and appreciate the comments from the representation learning point of view on inductive biases.
>
> **Role of inductive biases**
>
> We agree that inductive biases should be used sparingly, especially when large amounts of data are available, as is often the case for unsupervised methods. Slot Attention makes use of a fairly minimal set of inductive biases: permutation equivariance of functions operating on the objects or slots, and softmax normalization across slots (instead of per slot), i.e. the scene has to be separated into the available slots, which is a direct implementation of scene-object decomposition principles.
>
> In fact, our novel Slot Mixer relaxes the strong inductive bias of the RGB alpha blending in the popular Spatial Broadcast decoder, replacing it with a more powerful blending in feature space.
>
> Compared to prior art in 3D scene decomposition, OSRT makes fewer assumptions about the scene. Pixels are rendered directly instead of baking the physical volumetric rendering process into the model, or assuming a surface representation of the scene such as ObSuRF's depth supervision during training [27]. Furthermore, we do not explicitly address the background, for which several assumptions are made in uORF [34], e.g. the assumption that all slots should reside in a central scene bounding box.
>
>
> **Variance in synthetic datasets**
>
> We agree that synthetic datasets often miss "variance" that is found in real-world datasets, such as outliers, biases, and overall variety. That said, we evaluate OSRT on the very challenging MSN-H dataset. For example, in contrast to datasets of prior works (uORF [34], ObSuRF [27]), objects and backgrounds have textures, and the cameras are sampled fully randomly instead of having fixed positions. Importantly, the dataset was not specifically designed for scene decomposition, but rather proposed by SRT [23] as a test bed for non-trivial novel view synthesis.

---

### Author Response · Authors · 2022-08-02
**General response to reviews**

We thank all the reviewers for their positive feedback and helpful comments. In particular, we appreciate that the reviewers find the paper to be well-written (jvub, VX3v, HAVj), our method to be technically sound and intuitive (jvub, 3uqV, HAVj), and that they agree our method is much more efficient and significantly outperforms prior art in both image quality and scene decomposition (jvub, VX3v, 3uqV, HAVj), without requiring further supervision such as depth maps (3uqV). The reviewers further welcome our extensive experimental evaluation (jvub, VX3v, 3uqV), highlighting in particular the importance of the role of novel-view synthesis for scene decomposition (VX3v, 3uqV).

We address comments below in direct replies to the reviews.

---

### Meta-Review · Area_Chair_c1Dd · 2022-08-26

**Recommendation:** Accept
**Confidence:** Certain

**Metareview:**

The paper received positive leaning reviews (2x borderline accept, 1x weak accept, 1x accept). The meta-reviewer agrees with the reviewers' assessment of the paper.

**Award:**

No

---

### Decision · Program_Chairs · 2022-09-14

Accept